# Advancing Access to Healthcare through Telehealth: A Brownsville Community Assessment

**DOI:** 10.3390/healthcare10122509

**Published:** 2022-12-11

**Authors:** Edna Ely-Ledesma, Tiffany Champagne-Langabeer

**Affiliations:** 1Department of Planning and Landscape Architecture, University of Wisconsin-Madison, Madison, WI 53706, USA; 2School of Biomedical Informatics, University of Texas Health Science Center at Houston, Houston, TX 77030, USA

**Keywords:** telehealth, Latinx, community engagement, planning, public health

## Abstract

(1) Background: This paper focuses on the development of a community assessment for telehealth using an interprofessional lens, which sits at the intersection of public health and urban planning using multistakeholder input. The paper analyzes the process of designing and implementing a telemedicine plan for the City of Brownsville and its surrounding metros. (2) Methods: We employed an interprofessional approach to CBPR which assumed all stakeholders as equal partners alongside the researchers to uncover the most relevant and useful knowledge to inform the development of telehealth community assessment. (3) Results: Key findings include that: physicians do not have the technology, financial means, or staff to provide a comprehensive system for telemedicine; and due to language and literacy barriers, many patients are not able to use a web-based system of telemedicine. We also found that all stakeholders believe that telehealth is a convenient tool that has the capacity to increase patient access and care. (4) Conclusions: Ultimately, the use of an interprofessional community-based participatory research (CBPR) design allowed our team to bring together local knowledge with that of trained experts to advance the research efforts.

## 1. Introduction

Telehealth is a broad term that describes the remote provision of health care using technology such as telephone, apps, or web-based platforms [1,2]. Telemedicine is a subset of telehealth that refers specifically to the provision of clinical health care services ranging from asynchronous transmission of information or synchronous, live conferencing, between patient and clinician. For this assessment, we used the definition of telemedicine as defined by The Center for Connected Health Policy (CCHPCA) which states that telemedicine is “a collection of means or methods for enhancing health care, public health and health education delivery and support using telecommunications technologies. Telemedicine encompasses a broad variety of technologies and tactics to deliver virtual medical, health, and education services [3]”.

In a virtual visit, the patient and clinician are connected via a live, synchronous, interactive video system. Some researchers believe that making information, education, and management resources readily available to patients, telehealth allows individuals to become partners in their own health, thus empowering them to make decisions along-side their care providers [4,5]. Established standards for evaluating telehealth interventions recommend that a crucial time for evaluation of a telehealth plan is during the conceptualization and design phase [6]. Research shows, however, that many telehealth interventions are only evaluated at the end of the study period, once the intervention has been fully developed, tested, and implemented [4,7].

Although telehealth was originally used to access patients in remote locations, virtual visits have increasingly been accepted as a tool to provide real-time, convenient medical care across a wider range of geographies and can have significant cost–benefits, especially in prehospital environments [8,9,10,11]. The recent shift in use of telehealth can be attributed to two factors: (1) the rapid advances in technology and the widespread affordability and accessibility of basic telemedicine tools such as mobile devices [8,12]; and (2) the need to isolate and social distance due to the coronavirus disease (COVID-19) pandemic [13,14,15,16]. During the pandemic, telehealth emerged as a major force as insurers and the government changed reimbursement and other policies which made it easier for patients and providers to use virtual consultations with patients [17,18]. Thus, the pandemic offered the healthcare system an opportunity to increase access to healthcare through telehealth. 

This paper focuses on the development of a community assessment for telehealth using an interprofessional lens, which sits at the intersection of public health and urban planning using multistakeholder input. While studies have found that interprofessional interactions between public health and urban planning research is limited, there is a desire for greater collaboration between the fields [19]. Engaging directly with built environment issues is critical as disruptions to the health and urban systems are likely to reshape access barriers to care today with potential long term implications [20]. Little is known about how providers are currently using telehealth tools and ways that they may promote healthcare more effectively [21]. In addition to that, few studies have explored telehealth through recruitment methods that target racial and ethnic minority populations from resource-limited areas for community-based health and needs assessments [22]. We aim to fill that gap to champion the of a design of a community-based health and needs assessment in South Texas.

Community-based participatory research (CBPR) has been championed a methodological approach that can bring together local knowledge with that of trained experts to advance urban planning efforts [23]. In public health and healthcare research, CBPR principles are frequently used to engage at-risk communities in intervention development, with a community-based health and needs assessment being a first step to define priorities [24]. Nevertheless, very little is known about best practices for engaging individuals in community-based health and needs assessments that focus on telehealth [22]. To address this gap in the literature, we employed an interprofessional approach to CBPR which assumed all stakeholders as equal partners alongside the researchers to uncover the most relevant and useful knowledge to inform the development of telehealth community assessment [4].

## 2. Materials and Methods

Our main objective for the research was to create a plan for implementing telemedicine and connected health technologies broadly across the City of Brownsville and surrounding metropolitan areas. To complete this assessment, we assembled an interprofessional team of experts from the fields of urban planning, health technology policy, technology implementation, and telemedicine. This assessment took place between April and September of 2021 and was broken up into three phases: exploration, assessment, and utilization. 

During the exploration phase, we identified the users of this information, the geography of the defined coverage area, the demographic profile, the current internet capacity, and a list of internet providers. We defined the assets of the region, noting the number and genres of healthcare providers, universities, think-tanks, regional accolades, and unique capabilities. It was during the exploration phase that our disciplinary silos were bridged to design a research approach that encompassed both an urban planning and health technology policy approach to better inform the project assessment needs. Lastly, in this exploration phase, we reviewed and collate all relevant laws which apply to telemedicine, privacy and security, and healthcare reimbursement.

During this assessment phase, we sought contemporary feedback from stakeholders. Specific stakeholders included clinicians, patients, and leaders in the community. We used a variety of methods which included survey, focus groups, and one-on-one qualitative interviewing techniques. All methodologies were reviewed by the Institutional Review Board of the University of Texas Health Science Center. We also assessed acceptance factors to technology for both clinicians and patients, noting gaps and opportunities for success. 

During this phase, we sought feedback from stakeholders including clinicians, patients, and leaders in the community. 

One-on-one meetings were held in person with hospital leaders, outpatient clinics, physician offices, and healthcare care clinics.Individual focus groups were conducted with patients, interested parties, and Industries.Follow-up calls were made to multiple individuals until conceptual saturation.Anonymous surveys of clinicians, patients, and stakeholders were broadly distributed in English and Spanish through community networks and collected through a Qualtrics database.

For the in-person meetings, several provider organizations in the region gave us their time, expertise, experiences, and aspirations for telemedicine. The one-on-one meeting participants were identified through snowball sampling [25]. Participants for the one-on-one meetings were recruited through clinic and provider rosters provided by the local municipal partners. They were contacted via phone and email and were not compensated for their participation. 

These organizations that participated in the meetings included: Rio Grande Valley Health Information Exchange (RGVHIE)Physician Executive (practice owner), Chairman of local health organization, Executive Director, South Texas Physician AllianceSu Clinica, Federally Qualified Health Center, located in Brownsville and HarlingenProyecto Juan Diego, BrownsvillePhysician Executive (practice owner) and Brownsville Commissioner at LargeValley Baptist Hospital

We asked provider organizations to assess the potential for a telemedicine program to impact value specific to their revenue, health outcomes, and patient experience. We asked the following:Will the program require a significant expenditure for the provider?How will the program impact workflow?Does the provider organization have the technology to support an implementation?Will your patients be able to access and use the telemedicine services?What is on your wish list for implementing an ideal telemedicine program?

Meetings with providers were arranged by the project principal investigator one month in advance of our visit. Meetings were scheduled back-to-back over a three-day period on 14 June through 16 June 2021. The research team met with practitioners for one-hour sessions and used the questions above as a guide for the discussion. The purpose of these conversation was to gain insights into their experience with telehealth and to have them strategize on what defines an ideal telemedicine program for their needs. Conversations from the one-on-one meetings offer a standardized method for gathering information from multiple respondents, while allowing the flexibility to pursue interesting threads that may arise in conversations [20,26].

On Tuesday 28th June 2021, we held an in-person focus group at the Brownsville Chamber of Commerce. This focus group engaged with key community partners. Representatives from local non-profits organizations, and both the public and private sectors from across Cameron County were invited. Participants for the in-person focus group were recruited from a listing of local community partners prepared by the City of Brownsville. Participants were invited via email and asked to confirm their attendance. The focus group was conducted in at the local Chamber of Commerce as this was seen as a neutral location to discuss a regional planning strategy. The focus group was conducted in English by a member of our research team, and they were assisted by four student volunteers from the Texas Southmost College, a local community college. These students served as facilitators during our breakout sessions. The focus group began with an introductory presentation to the topic by our research team. The focus group questions were presented and then the participants were invited to divide into two smaller discussion groups to facilitate more candid discussions. Two students were assigned to each group; one student facilitated the discussion while the other took notes. The discussion sessions ran for an hour then the groups reported back to the larger group for a thirty-minute discussion. 

Organizations and entities that attended included:City of Brownsville Public Health DepartmentCity of Brownsville EMS UnitCameron County Health DepartmentBrownsville Chamber of CommerceUniversity of Texas Health Science Center at HoustonTexas Southmost CollegeBrownsville Independent School DistrictBrownsville Wellness CoalitionIndustry

The focus group addressed the following questions:Question 1: What is your experience with telehealth, and how does it impact your company or organization?Question 2: What do you think is the relevance or value of telehealth to your company or organization?Question 3: What obstacles does your company or organization face in providing telehealth and or healthcare to employees?Question 4: If given the option, list reasons for why you would (or would not) chose for your company or organization to participate in a regional telehealth plan?

Two in-person focus groups were held on 15 and 16 July 2021 targeting patients. The focus groups were held at Bob Clark Social Service Center and the Proyecto Juan Diego, both locations for community resources located in low-income areas in Brownsville. Sessions were held in both English and Spanish. Participants were recruited from patients currently participating in a parallel study by the University of Texas Health Science Center at Houston. The focus group centered on gaging patient buy on with utilizing telehealth. Patient focus groups were conducted via Zoom. They were conducted in Spanish and facilitated by a researcher from the University of Texas Health Science Center at Houston who has a working relationship with the participants through other ongoing studies. The patient focus groups were supported by a Texas Community College student volunteer that served as support for the moderator and notetaker. Each session last one hour. 

Audio recordings of the focus group were transcribed verbatim. All Spanish-language transcripts were translated into English by a researcher fluent in both Spanish and English. Using the flexible coding method [27], the text was divided into larger sections with broader structural codes; these sections were then further parsed using more granular, conceptual and thematic codes. This approach allows for a more focused analysis of subsections of the data, which is particularly effective for a data set that will be used for multiple research groups [20]. The team used a qualitative descriptive approach to data analysis, identifying themes inductively and thematically. Qualitative description (QD) is often used in health research to inform the development of interventions or policies that can improve health outcomes for various populations [28]. On the basis of exploring “the who, what, and where of events and experiences”, QD provides a straight description based on participants’ responses, making use of participants’ own language to support the themes that emerge [4,29,30].

To overcome barrier to improve of health outcomes, it is important that researchers utilize practices that consider the social and cultural aspects of the population they intend to study [24]. Our diverse research team consisted of researchers native to South Texas, and to overcome challenges of researchers being viewed as outsiders, local community college students were hired to facilitate discussions at all focus groups. 

Once data was gathered through exploration and assessment, we proceeded to the final stage of utilization—where we made recommendations. These include a list of the health technology priorities based upon what the study finds. A list of technology recommendations was established; and from these, we created a set of use- cases. In implementation science, we create a “use case” to show how technology may be used in a variety of scenarios to provide a representation of a future state. We provided suggestions of potential pilots that can be tested in smaller areas to establish the feasibility of a broader implementation. 

## 3. Results

The following section provides a summary of the results of our telehealth assessment for the region. Overall, both the infrastructure assessment, which includes an analysis of the provider landscape and intellectual resources, and the provider landscape assessment focus on findings at the regional scale. The third portion of the assessment, the focus groups for community partners and participants, is focused mostly on the City of Brownsville. 

### 3.1. Infrastructure Assesment

The study took place in the Rio Grande Valley (RGV). The RGV is the southernmost region of Texas, consisting of Cameron, Hidalgo, Starr, and Willacy County. Cameron County is the southernmost county in Texas. The United States Census of 2019 estimated 423,163 people, of which an estimated 23% are foreign-born [31]. The majority population in Cameron County consist of people who identify as Hispanic (90%) [32]. Based on age, 29.9% of the population in Cameron County are younger than 18 years of age, and 13.8% are 65 years of age or older. Only 17.3% of the population 25 years and older have a bachelor’s degree or higher. According to the 2019 Census, it is estimated that more than one quarter of the population (25.5%) live in poverty [31]. 

Brownsville is the largest metropolitan city within Cameron County and located in the Rio Grande Valley. Known as the southernmost point of Texas, Brownsville sits adjacent to Matamoros, Mexico and has a growing population of 182,781 people [33]. The City of Brownsville is also home to the rural Cameron Park area, known locally as a *colonia*. The Spanish term *colonias* is used to describe unincorporated settlements, neighborhoods, or communities along the U.S. border with Mexico. These areas typically lack multiple elements of infrastructure commonly found in developed neighborhood such as paved roads, sewer systems, electricity, gas, and potable water [34]. 

Brownsville is one of the most impoverished metropolitan areas in the United States, where 25% of the population and 48% percent of children live in poverty, 30% of the population is uninsured, 80% of the population is obese or overweight, and 30% have diabetes with 50% of them unaware of it [35]. South Texas represents about 18% of the state’s entire population, of which more than 2/3 are Hispanic. The population has a low post-secondary education rate of only about 18%. Only about half the population has access to broadband internet. South Texas residents are confronted with poor health outcomes and health gaps compared to the state of Texas as a whole; these include tuberculosis, chlamydia, cancer, birth defects, diabetes, obesity, and lead poisoning.

Obesity and diabetes are endemic to South Texas, with incidence rates much higher than state and national levels. Obesity is a causal risk factor for diabetes and is directly linked to lifestyle behaviors, physical behaviors, and eating habits. Lower-income individuals and patients who do not have health insurance have a significantly higher likelihood of having undiagnosed diabetes, and the resulting costs in both economic and human terms can be devastating [36]. 

One overarching consideration here is that the Rio Grande Valley, being a border area, has a large number of undocumented residents who are more likely to live in poverty, have no health insurance and little education, and are reluctant to participate in US Census surveys due to fear of deportation. In recent years, the U.S./Mexico border has seen an increase of homelessness, domestic violence, and an increased number of individuals with substance use disorder in many areas in the region [37]. These factors all contribute to a population that lives in the shadows, is unable to receive government help with healthcare, and because of fears of the government and deportation, often do not get the healthcare assistance they need.

As noted above, about 30% of the population is uninsured. Of the 70% that are insured, 29.1% are on an employee health care plan, 27.4% are on Medicaid, 7.51% on Medicare, 7.17% on non-group plans, and 1.23% are on military Veterans Affairs plans [38]. There is still a large gap between insured and uninsured, contributing to unfavorable health outcomes. The need to improve health care coverage in this area has been an ongoing challenge. According to the Texas Medical Association, the uninsured are a diverse group that can or cannot afford private insurance and thus chose to not purchase it.

Current health care coverage options in the RGV are private insurance, government health care coverage such as Medicaid, children’s health insurance plan (CHIP), and Medicare. Governmental health care coverage programs require individuals to meet requirements to receive coverage, as well as frequent re-enrollments. This process can be tedious and difficult for individuals to understand, leading to eligible individuals not being enrolled. 

For the uninsured, there are different ways in which they might seek healthcare. In Cameron County, several Federally Qualified Health Centers (FQHCs) community-based health care providers receive funds from the HRSA Health Center Program to provide low-cost to no-cost primary care services to qualifying individuals. Cameron County Public Health has a clinic in Harlingen, San Benito, Brownsville, and Port Isabel. Cameron County Public Health also offers an Indigent Health Program for county residents at or below 21% of the federal poverty line, with resources less than $2000, who do not qualify for other state or federal healthcare programs such as Medicaid. Indigent Health Care provides medical screenings, annual physical examinations, inpatient and outpatient hospital visits, and laboratory and radiology services [39]. These disparities are exacerbated in residents living in the most rural areas where they have a greater risk of disease and substance abuse. The secondary effect of living without clean water and the overall lack of infrastructure for these residents puts them at a higher risk of asthma and environmental allergies [40]. 

#### 3.1.1. Provider Landscape

The Health Resources and Services Administration (HRSA) declared Cameron County to be a Health Professional Shortage Area (HPSAs) and a Medically Underserved Area (MUA). MUA are geographic areas and populations with a lack of access to primary care services. HPSAs are designations that indicate health care provider shortages in primary, dental health, or mental health. 

Cameron County has a Local Health Department, Cameron County Public Health (CCPH), and the City of Brownsville (COB) also has a health program to assist in addressing and serving community health needs. In addition to local government resources, other institutions in the community provide healthcare and assist in educating patients about healthy lifestyle choices. The Rio Grande State Center in Harlingen is funded by the State of Texas and offers both in-house adult psychiatry services and outpatient services, including primary care, women’s health, and prescription assistance. The non-profit Proyecto Juan Diego targets low-income families through their educational programs and family activities with the aim to create community members who are self-sufficient and prioritize preventive health services. The project provides educational programs, family activities, advocacy, and preventative health services. During the focus group, a University of Texas Health Science Center researcher stated, “the people that we work with, they’re not used to the healthcare system working for them.” Figure 1 shows the locations of healthcare providers in Brownsville. 

Brownsville and the RGV are home to rural *colonias*, economically distressed high poverty communities lacking one or more essential community infrastructure elements such as paved roads, sewer and water systems, electricity, gas, and most importantly for this report, health services. Cameron Park in Cameron County is a prime example of a *colonia*, with a population that is 98.7% Hispanic. In *colonias*, *Promotoras* are health educators who work within the community to educate residents on various chronic health conditions and preventive measures targeting obesity, diabetes, maternal health, breastfeeding, and health care access information. 

Mexico offers affordable and easy access to health care services such as prescription medications and services from dentists and doctors. Health Care Utilization, a study conducted on border counties, revealed that individuals who went to Mexico for health care needs had similar characteristics: the majority were Hispanic and spoke Spanish, and were already familiar with health care services in Mexico; about half were low income or at the poverty level, 47% were uninsured, and 10% expressed dissatisfaction with health care services in the U.S. side [41]. For many, healthcare in Mexico is not preventative but rather is obtained when illness strikes; as a result, this option does little to prevent or mitigate underlying risk factors of chronic diseases such as diabetes and hypertension.

#### 3.1.2. Intellectual Resources

In Cameron County, there are nine postsecondary education academic intuitions; seven are private and three public schools. The University of Texas Rio Grande Valley (UTRGV) is a four-year, public university which offers 293 academic programs. UTRGV is the largest postsecondary institution in the RGV, with undergraduate and graduate school enrollments totaling nearly 30,000 students, with 89% being Hispanic. The UTHealth Rio Grande Valley School of Medicine began enrollment of students after accreditation in 2015, and now currently enrolls over 200 medical students along with over 200 medical residents in 16 accredited residency programs including family medicine, internal medicine, obstetrics and gynecology, and psychiatry. 

The University of Texas Health Science Center (UT Health) at Houston has a satellite campus in Brownsville for the School of Public Health and the School of Biomedical Informatics, offering graduate and doctoral level programs to dozens of students annually. 

Texas Southmost College (TSC) in Brownsville is a public junior college that offers the first two years of education towards a bachelor’s degree, as well as certificate programs, associate degrees, and technical education. Texas State Technical College (TSTC) in Harlingen is a public college that offers 171 programs.

### 3.2. Provider Assessment

The provider survey had 33 respondents including clinicians and executives with 26 coming from physician practices. Figure 2 shows a breakdown of the distinct roles of the 33 respondents. 

Key findings of the survey include:The primary use for telemedicine at this time is primary care followed by e-prescribing.Physicians do not have the technology, financial means, or staff to provide a comprehensive system for telemedicine.Due to language and literacy barriers, many patients are not able to use a web-based system of telemedicine.Many patients do not have a computer and their broadband is limited.Patients prefer to see their provider in person but would use the telephone to communicate healthcare issues with their physician.For both the provider and the patient, the use of a phone without video provides the most reliable tool to access remote care, but its utility for telemedicine is very limited.

#### 3.2.1. Physical Barriers to Telemedicine

The key barriers include lack of knowledge to implement an effective telemedicine program. Other concerns include lack of technology to support telemedicine, staffing and workflow concerns. Lack of funding and a clear understanding of the reimbursement policies for telemedicine are also listed as barriers.

All the providers believe that a significant barrier to a successful telemedicine program is that patients do not have adequate resources—both financially and technically—to use telemedicine services. Figure 3 shows a breakdown of the technology-based barriers to telemedicine based on the provider survey. This was confirmed through patient focus groups. Although providers felt that they had an adequate system to educate patients about telemedicine offering, few of the patients we spoke to know their providers had this service.

Other concerns include lack of knowledge for appropriate codes for different types of telemedicine visits, so visits are reimbursed appropriately, and clear guidelines and standards are met.

#### 3.2.2. Physical Drivers of Telemedicine

When asked if telemedicine can help patients manage their health, 47% of the providers strongly agreed with this statement, particularly as they believe that the key reason patients had difficulty accessing care is due to lack of transportation. Providers also mentioned that their patients desired after hours medical care which can be provided by telemedicine.

For many providers, telemedicine was not utilized prior to the pandemic. There were concerns of how to implement the technology, how it would impact workflow, and how their patients would be able to utilize the system. One physician state, “Telemedicine would streamline the process, and physicians could see more patients.”

There was consensus among all those we met that using telemedicine minimizes COVID-19 risk to healthcare workers and patients. The providers can screen patients remotely rather than having them visit the practice or hospital and deliver care for those who do not need medical intervention or can receive care at home. Providers are also able to proactively communicate with their patients and use telemedicine for after-hours access. Another physician stated, “Telemedicine is fabulous for a screening tool to determine if a clinic visit is needed.”

Other benefits include having access to additional providers, including specialty providers. Su Clinica has only 3 physicians on staff but can access other providers remotely to increase access to care and provide additional services such as specialty care, behavioral health, patient education, and pediatric care while improving work efficiency and helping to meet clinical outcomes.

### 3.3. Focus Group Findings

#### 3.3.1. Community Partners’ Experience with Telehealth

Key takeaways from the discussion were that while telehealth is an option for employees in most of the represented companies, workflows are ambiguous for patient buy-in, staffing is limited, and there are questions about funding moving forward.

“*We recently had our annual health fair, and our insurance company contracted someone from San Antonio at another clinic… we set up the computers, we set up everything for them [employees], we told them just sit here and wait until the nurse connects with you. But otherwise, if we would have asked, just connect here, they [employees] wouldn’t have done it*.”—Industry, CEO

Of the organizations present, the following provide health care that includes access to telehealth for their employees: City of Brownsville, University of Texas School Health Science Center at Houston, Brownsville Independent School District, and a large local Industry. A representative of the Brownsville Fire Department stated that even though they have the technology, the city lacks both connectivity and the platform to run telehealth.

While those present are community leaders in the area, a prevailing theme across the focus group participants is that most of their employees are not using the telehealth services that they indeed have access to. In the case of the large Industry employer, a private sector company, they have been offering telehealth to their employees for serval years, but most did not start to use it until they were forced to during the COVID-19 pandemic. The University of Texas Health Science Center at Houston transitioned to a virtual telehealth model during the COVID-19 pandemic to consult with over five thousand patients in their Chronic Disease Management Program. Of the nine thousand, nearly 25% have no access to the internet at home nor on a smart phone/devise. Those that do (75%) needed assistance to connect to the internet.

#### 3.3.2. Community Partners’ Perception of the Relevance of Telehealth

Key takeaways on the relevance of telehealth are that telehealth is convenient; it can increase capacity for patient access, and could increase access to specialty care, specifically mental health services.

“*Our students, most of the kids, I would say at least 70% or 80%, the closest to healthcare they have is contact with the nurses on campus. So that’s what telehealth could help with*.”—Brownsville Independent School District (BISD), Administrator

A recurring theme across the focus group participants is the need to understand the health care context of Brownsville and the South Texas region. As a region with high rates of uninsured residents, access to healthcare is limited for many patients. 

“*The biggest benefit is having the right care for the right patient, at the right time*.”—Brownsville EMS, Chief

The community health clinics which see a high volume of uninsured residents have long wait lists and lack capacity to service all those in need. If a telehealth model were introduced in Brownsville, it would allow health care providers to increase capacity and reach a wider net of patients. Increased access and more frequent doctor visits could have potential long-term benefits to decrease health disparities in the region. 

“*It [telehealth] can be so valuable if people can be made to feel comfortable with it, and they have internet access. Obviously if they [patients] can’t access it, they [doctors] can’t make them access it*.”—The University of Texas Health Science Center in Houston, Researcher

#### 3.3.3. Community Partners’ Perception of Obstacles for Telehealth

Key obstacles for telehealth in Brownsville noted by the focus group participants were the language barrier for Spanish speaking population, trust issues which might limit buy-in for patients, the need to educate users on the value of the service, cultural stigmas, and the lack of resources to implement the technology.

To increase access to telehealth it is important that any service adapts to the needs of the local population. Starting with language, Spanish-only households face an increased burden in navigating virtual platforms available in English only. In addition to that, there is the obstacle of limited technological skills of the older population and thus there is a need to address how to provide basic computer skills to this group through some type of telehealth educational plan. Among the focus group participants, a common theme was potential users were skeptical of the value of telehealth. For example, for the large Industry employer, even though the company offers their employees an incentive of reduced health insurance premiums for using their insurance telehealth plan, employees remain hesitant.

“*Even though it’s a privilege to have all these things [telehealth], people do not give them the importance that it has*.”—Industry, CEO

A common thread among the focus group participants was the possible attribution of cultural stigmas of healthcare in general. To address this limitation, the Brownsville Wellness Coalition suggested the use of peer support to increase buy-in. This could be achieved through community leaders or local ambassadors willing to champion a campaign to increase buy-in.

“*If we find some leaders of the community itself and if that person can relate to that person too… when it comes from your employer or your boss, it kind of feels like it’s direct, you know, it’s forced. Pero si la comadre te invita, y que vamos…it’s always that way. That’s also a kind of incentive*.”—Brownsville Wellness Coalition, Director

From the public sector side, entities such as Cameron County and the City of Brownsville expressed concerns on the limited internet capacity. Government employees lack bandwidth and often struggle to find the resources to provide telehealth services to a wider segment of the population. In 2019 Brownsville EMS began a federal pilot program called ET3 (treat, triage, and transport). The primary purpose of this program is to reduce the number of people that use the emergency room as their primary health care. One alternative that the program outlines is to provide an alternative destination such as a 24 h urgent care site which Brownsville currently lacks. A second option is telehealth, but there is no plan currently in place to implement this for the city.

Within the private sector, a shift to telehealth might see pushback from hospitals and private practices as this might imply a loss of revenue due to the standard billing practices for in-person consultations versus virtual. A challenge will be to increase doctor buy-in from the private sector for the hesitant.

#### 3.3.4. Patient Drivers to Telemedicine

Of the 13 patient focus group participants, 3 had graduated high school, 2 spoke both English and Spanish, 8 had no insurance, and 11 had Smart Phones. The responses from the patient focus groups provided a wide range of responses to telemedicine—from a patient not knowing what telemedicine is to another patient having a great experience. Most of their telemedicine experiences occurred during the pandemic and the clinical sessions were conducted over the phone—not on a computer using video.

For patients with limited access to transportation, telemedicine provides an opportunity for them to connect with their providers by overcoming a key barrier to their accessing healthcare. Other patients would use the option of a telemedicine session, but ultimately prefer to see their provider in person.

Other comments from the patient focus groups include:Younger patients are more tech-savvy than older patients and thus more likely to utilize and benefit from telemedicine.Everyone has a smartphone, but they often use only the most basic features.Telemedicine can provide greater access to specialists out-of-town.Diabetic patients like the possibility of providing their blood or sugar levels to their provider remotely.Many patients would be willing to try telemedicine.Patients are happy with the program overall at *Proyecto Juan Diego*.The patients were more enthusiastic about a hybrid model, providing a balance between Telemedicine and in-person visits.

“*It is a great option for follow up as it saves time just to be able to let the doctor know your child is doing better*.”—Community patient

“*Telemedicine is probably ok for pediatrics, but it is a lot more important to go in person or go for more serious occasions*.”—Community patient

#### 3.3.5. Patient Barriers to Telemedicine

Some patients cannot use telemedicine as they are cash only and cannot be billed to insurance for the provider services. Figure 4 displays the most common responses provided as to why patients have difficulty accessing care.

The technical barriers include the fact that many of the focus group participants had no computer or tablet, either lacked internet or if they did it was very slow, and they do not feel confident learning about technology. Poor cellphone coverage resulting in dropped calls was a challenge for some and having more than one person online is difficult with limited broadband. Some of the patients preferred using a phone over a video as they felt it invaded their privacy. Other barriers include illiteracy in both English and Spanish so that for some, navigating a website would be difficult if not impossible.

“*The signal in the Valley is not very good, so you lose connection, if you want a better cell connection, you will have to pay way more and it is not always beneficial*.”—Community patient

## 4. Discussion: Lessons from an Interprofessional Approach to Developing a Community Assessment for Telehealth

Brownsville, a community with high health risk indicators, is place that could benefit from the long-term implications of a regional telehealth strategy. Through the interprofessional study, we were able to glean insights into the nuances of implementing such strategy both from the perspective of providers and the community at large. The infrastructure assessment demonstrated that there is a diversity of health care resources across the region, but there is a need to further connect these through community partnerships. Studies show that characteristics of the built environment can be determinants of the level of adoption of telehealth since living in certain types of environments may favor performing activities from distance [42,43,44]. Links between the built environment and telehealth might in turn influence how cities are built or adapted, and whether and how residents travel to access healthcare resources [44,45,46]. Overall, our research findings reinforce those of previous studies which illustrate that telehealth has the potential to increasing access for continuous care in rural areas and increase access to patients that lack transportation to health care facilities [20,47]. Studies show that individuals living in rural areas, racial and ethnic minority groups, and the elderly face higher rates of transportation barriers to care leading to poorer health outcomes and worsening of chronic conditions [48].

Nevertheless, while interventions involving telehealth technology show promise in promoting health care engagement in communities lacking health infrastructure [21], a key finding across our various stakeholder groups was the need to address context-specific barriers to buy-in. Similar to other studies, assistance with technology gaps would be key to a successful deployment of regional telehealth plan [49]. The study shows that both patients and providers see a complex interplay of patient-level barriers to access, such as individual interest and technology access, in addition to macro-level barriers to access, such as software access, funding, and personnel [49,50].

The community assessment gave providers an opportunity to strategize and outline priorities in developing a regional telehealth strategy. Providers were therefore asked to list their wish list is for the plan. Common responses included:Additional support and education for diabetic patients.Flexible work shifts: several mentioned using a Hybrid Model (e.g., one visit in person followed by a telemedicine visit).Provide resources to address literacy and basic computer skills.Add behavioral health to telemedicine due to low numbers of behavioral health providers.Ability to take vitals, labs through telemedicine.

“*Telemedicine provides seamless communication with providers regardless of where they are*.”—physician

The focus group participants [see Figure 5] also highlighted several key considerations when developing a regional telehealth plan: there is a need to integrate measures of accountability, regional approaches must engage with rural *colonias*, a hybrid option would make the system more robust, and citizen representation is essential for equitable engagement. Research suggests that culturally tailored interventions can lead to enhanced treatment engagement and improved treatment effectiveness [51,52]. What some might acceptable is contextualized and interlinked with prevailing social and cultural norms, therefore understanding and designing for such norms would therefore be critical to a successful plan implementation [53]

“*Our healthcare system is so uncoordinated that it [a regional telehealth plan] could help coordinate our healthcare system and people could have their information consistent*.”—University of Texas Health Science Center in Houston, Researcher

“*A regional plan is a good idea, but if it’s being set up to benefit the many… We just have to make sure that regional really means ‘region that benefits everyone’… make sure we have the same goal…I think anytime you don’t want to regionalize, we’re hurting ourselves because the region itself is a powerful voice… so we need to make sure that our voice is loud enough to sit on those tables and those conversations, city, county and so forth and saying, ‘Hey, we really need mental health’. That [our] voice is always a part of it*.”—City of Brownsville Public Health Department, Director

Generally, all participants agreed that developing a regional telehealth strategy or plan is a good idea. A key benefit outlined was that a regional plan could increase access to healthcare to a wider range of people across the RGV. There is an opportunity to increase outreach into areas of the valley with some of the most marginalized groups such as rural residents in *colonias.* The City of Brownsville could collaborate with other municipalities and regional governing bodies to define what scale of regionalism is appropriate in defining a regional telehealth plan. To ensure that a regional telehealth plan is successful, recognizing the need for a hybrid approach is critical. Rural residents need access to general consultations but identifying strategies to integrate lab visits and physical exams would be necessary. The role of the *promotoras*, the healthcare workforce that serve as a bridge between provider and patient, could be further leveraged as a conduit between the patients and new technology. Training the existing workforce on telemedicine utilization may also motivate the population to try new technologies. Indeed, *promotoras* have been the catalysts in many public health initiatives where new technologies were successfully implemented in technology-naïve population [54,55].

Participants emphasized the need to ensure accountability for any third-party agency or company that might operate a regional telehealth plan. Some expressed concerns about a doctor-provider-driven plan, which might marginalize representation from the public sector or the community. To address this concern, two recommendations were made. First, contractors must be held accountable to a minimum threshold of quality measures. Second, these measures would need to be defined by a local governing board. A governing board could be configured with citizen representation. Two successful models currently operating in Brownsville that the city could replicate are the boards for the Brownsville Housing Authority and the Proyecto Juan Diego.

## 5. Conclusions

This study applied an interprofessional lens to explore the development of a telehealth plan for the Brownsville, Texas. The collaboration between experts in the fields of urban planning, health technology policy, technology implementation, and telemedicine allowed for a more holistic approach to the research design. This study directly informed the design of a regional telehealth plan. We distilled and highlighted internal and external forces impact the community and outlined potential health information technology project implementation strategies. These strategies applied a tiered approach to model implementation based on infrastructure and human capacity. The intersection of needs would not have been so readily identifiable had it not been for the interprofessional approach to the research. Furthermore, these approached would go on to inform policy recommendations to implement a pilot study for implementing a telehealth plan in the region.

Our project aimed at addressing concerns in the literature which suggest that a crucial time for the evaluation of a telehealth plan is during the conceptualization and design phase, not post-evolution [56]. To address this concern, our team was able to leverage the capacity and expertise of the project partners by allowing each researcher to apply research methodologies that their fields deemed appropriate for the evaluation of the various components of the study. By doing so, we captured the needs of the region through the infrastructure assessment, then were able to triangulate those findings with those extracted from the provider surveys, and the in-depth discussion help with the key stakeholder and patient focus groups. The interprofessional community-based participatory research (CBPR) design allowed our team to bring together local knowledge with that of trained experts to advance the research efforts [23]. Further engagement with the community would be needed to make the process more robust, this would require additional touchpoints or feedback loops to continue to engage the community in the expansion of development of any city driven plans.

## Figures and Tables

**Figure 1 healthcare-10-02509-f001:**
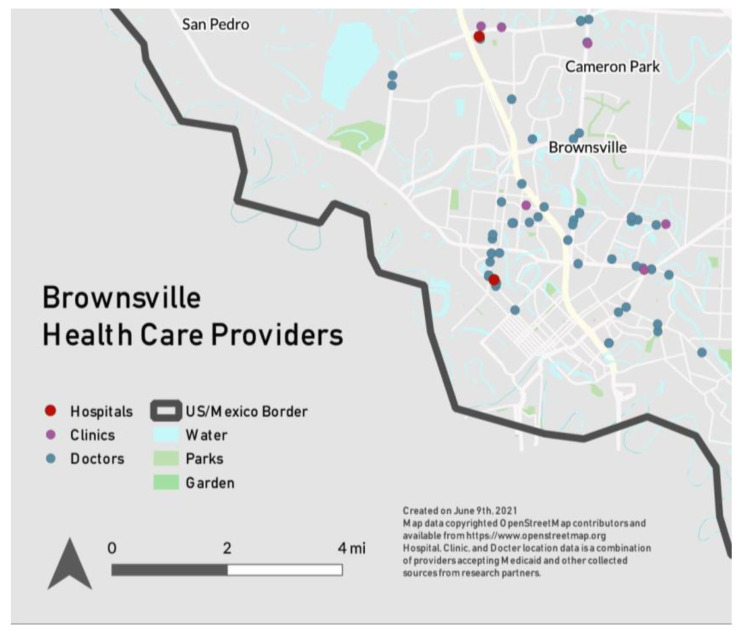
Map of health care providers in Brownsville, Texas.

**Figure 2 healthcare-10-02509-f002:**
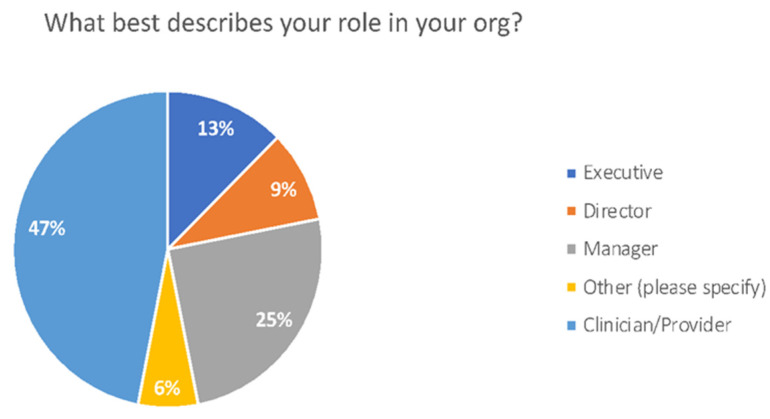
Role of providers survey.

**Figure 3 healthcare-10-02509-f003:**
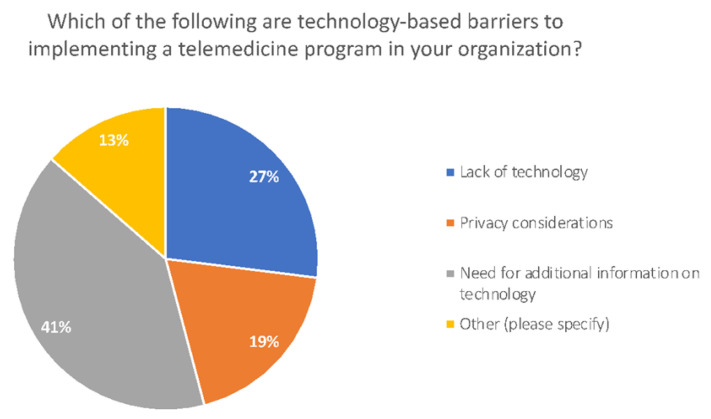
Technology-based barriers to implement telemedicine.

**Figure 4 healthcare-10-02509-f004:**
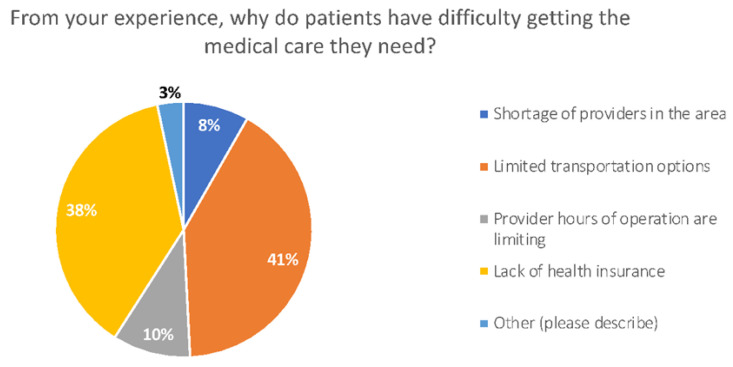
Barriers to medical care for patients.

**Figure 5 healthcare-10-02509-f005:**
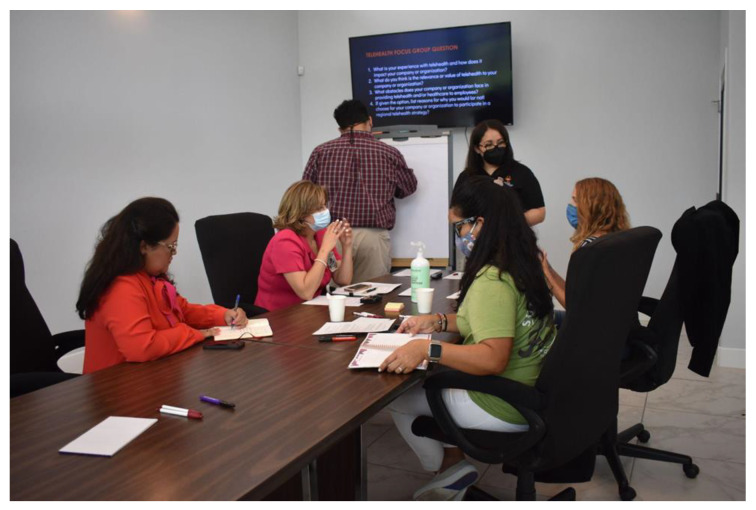
Photo of community partners focus group.

## Data Availability

Not applicable.

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
