# Peer review of "Advancing Access to Healthcare through Telehealth: A Brownsville Community Assessment"

_healthcare, 2022, doi:10.3390/healthcare10122509_

Round 1

Reviewer 1 Report

 The authors do touch upon a relevant topic, and I’m positive about the way they tried to ground their approach on literature.

However, most of the paper is devoted to a rather straightforward description of the results. No matter how important this is for the region that is subject of this research regarding needs in (tele-)healthcare, the strict results are not always interesting or transferrable to the context of possible readers of this paper.

Therefore, I would suggest to shift the focus a little bit in some sections, which means to highlight and motivate the approach and methodological decisions that were taken. That is information that might be used by other researchers / practitioners / governmental stakeholders in their own context of use.

In particular I’d suggest the following adaptations.

In section 2 on Materials and Methods, the authors mention three phases: exploration, assessment and utilization. The general meaning of the terms/stages is well explained. However, in the more elaborate descriptions, it is not always clear when it is about exploration / assessment /utilization. I’d suggest to have a look at this, and try to make it very explicit as well as give similar attention to the stages.

Please focus more on details of the methods such as recruitment, how the focus groups / interviews were exactly done, role of facilitators/moderators, prepared materials, the length of the sessions, etc, and similar for interviews. It is exactly this methodological information that would be interesting for other readers.

The discussion section does a good job in linking the results to general findings/literature. I think it should however be strengthened to accept this paper. This can be done in two ways: (1) elaborate on the next steps of the initiative that build on these results and (2) increase even references to literature, to other contexts/environments with similar findings etc.

Furthermore, these are some language errors / sentence constructs that not seem write/ typos:

Line 11: analyzes

Line 69: championed a -> as a?? (not sure)

Line 136: “held” is 2x in sentence

Line 181: sentence construct?

Line 188: findings -> finds?

Author Response

In section 2 on Materials and Methods, the authors mention three phases: exploration, assessment and utilization. The general meaning of the terms/stages is well explained. However, in the more elaborate descriptions, it is not always clear when it is about exploration / assessment /utilization. I’d suggest to have a look at this, and try to make it very explicit as well as give similar attention to the stages.

Point 1: Please focus more on details of the methods such as recruitment, how the focus groups / interviews were exactly done, role of facilitators/moderators, prepared materials, the length of the sessions, etc, and similar for interviews. It is exactly this methodological information that would be interesting for other readers.

Authors’ Response 1: We have significantly expanded the methodology for the recruitment and the role of the facilitators in this section. We also clarified the framing of exploration/assessment/utilization within the methodology section.

Point 2: The discussion section does a good job in linking the results to general findings/literature. I think it should however be strengthened to accept this paper. This can be done in two ways: (1) elaborate on the next steps of the initiative that build on these results and (2) increase even references to literature, to other contexts/environments with similar findings etc. 

Authors’ Response 2: We have added to the discussion to include the role of the Community Health Worker (promotora) in the implementation of new technologies. This was added along with new citations.

Point 3: Furthermore, these are some language errors / sentence constructs that not seem write/ typos: Line 11: analyzes; Line 69: championed a -> as a?? (not sure); Line 136: “held” is 2x in sentence; Line 181: sentence construct?; Line 188: findings -> finds?

Authors’ Response 3:

All areas have been addressed in the manuscript.

Reviewer 2 Report

I believe that all studies on promoting access to health care through telehealth are very valid, even more so when they are limited to a community.

 To improve the manuscript, authors should explain in detail:

1. Clarify why they consider their results to be at the intersection of public health and urban planning. This, because they do not clarify public policies, nor government plans (nor of the community).

2. Clarify how the results differ for participants from the City of Brownsville and participants from its surrounding metropolitan areas.

3. Delimit the extent to which participatory research based on the interprofessional community (CBPR) remains, this because in the end solutions or proposals for consensus with the community are not proposed.

4. Update the following references, considered old:

- Ref: 2 and 49 of 2001

- Ref 5 of 1999

- Ref 21 and 44 of 2005

- Ref 24 of 2000

- Ref 47 of 2006

- Ref 32 and 33 of 2007

Among other.

Author Response

  1. Clarify why they consider their results to be at the intersection of public health and urban planning. This, because they do not clarify public policies, nor government plans (nor of the community).

Response 1: We clarified the intersectional approach of the research in the Conclusion.

  1. Clarify how the results differ for participants from the City of Brownsville and participants from its surrounding metropolitan areas.

Response 2: The distinction is clarified with a summary at the top of the Results section.

  1. Delimit the extent to which participatory research based on the interprofessional community (CBPR) remains, this because in the end solutions or proposals for consensus with the community are not proposed.

Response 3: This limitation is addressed in the Conclusion.

  1. Update the following references, considered old:

- Ref: 2 and 49 of 2001

Updated.

- Ref 5 of 1999

Updated.

- Ref 21 and 44 of 2005

Updated.

- Ref 24 of 2000

Updated.

- Ref 47 of 2006

Updated.

- Ref 32 and 33 of 2007

Updated.